Transformative insights from transcriptome analysis of colorectal cancer patient tissues: identification of four key prognostic genes

http://orcid.org/0000-0001-9353-9387 Belder Nevin 1
Charyyeva Sulgun 1
Abaci Oruc Edibe Ece 1
http://orcid.org/0000-0002-1398-637X Kawalya Hakiimu 1
Sahar Namood-e 1 2
Omidvar Nader 3
Savas Berna 4
Ensari Arzu 4
http://orcid.org/0000-0001-7940-2499 Ozdag Hilal 1 hilalozdag@gmail.com
1 Ankara University Biotechnology Institute , Ankara , Turkey
2 Comsats University, Institute of Information Technology , Islamabad , Pakistan
3 Cardiff University, Systems Immunity Research Institute and Division of Infection and Immunity , Cardiff , United Kingdom
4 Ankara University Faculty of Medicine, Department of Medical Pathology , Ankara , Turkey
Mitsouras Katherine
Electronic publication date: 2025 Aug 20
Publication date: 2025
Volume: 13
Electronic Location ID: e19852
Received 2024 Dec 10; Accepted 2025 Jul 15
Copyright: © 2025 Belder et al.
Copyright year: 2025
Copyright holder: Belder et al.
License: This is an open access article distributed under the terms of the Creative Commons Attribution License, which permits unrestricted use, distribution, reproduction and adaptation in any medium and for any purpose provided that it is properly attributed. For attribution, the original author(s), title, publication source (PeerJ) and either DOI or URL of the article must be cited.
License URL: https://creativecommons.org/licenses/by/4.0/

Keywords: Colorectal cancer, Transcriptome analysis, Biomarker, Prognostic, Risk model

Funding: Scientific and Technological Research Council of Turkey TUBITAK 109S477 Ankara University BAP 19L0415002, BAP 20B0415001 and TUBİTAK BİDEB 2216 This study was supported by research grants from the Scientific and Technological Research Council of Turkey (TUBITAK; Grant No. 109S477, Ankara University BAP 19L0415002, Ankara University BAP 20B0415001, TUBİTAK BİDEB 2216). The funders had no role in study design, data collection and analysis, decision to publish, or preparation of the manuscript.

==============================
Colorectal cancer (CRC) is a leading cause of cancer-related deaths worldwide, necessitating accurate and robust predictive approaches to assist oncologists with prognosis prediction and therapeutic decision-making in clinical practice. Here, we aimed to identify key genes involved in colorectal cancer pathology and develop a model for prognosis prediction and guide therapeutic decisions in CRC patients. We profiled 49 matched tumour and normal formalin-fixed paraffin-embedded (FFPE) samples using Affymetrix HGU133-X3P arrays and identified 845 differentially expressed genes (FDR ≤ 0.001, fold change ≥2), predominantly enriched in the extracellular matrix (ECM)-receptor interaction pathway. The integrative analysis of our data with publicly available mRNA and miRNA datasets, including their differentially expressed gene analyses, identified four overexpressed genes in the ECM-receptor interaction pathway as key regulators of human CRC development and progression. These four genes were independently validated for their differential expression and association with prognosis in a newly collected CRC cohort and publicly available datasets. A prognostic risk score was developed using these genes, with patient stages weighted by multivariate Cox regression coefficients to stratify patients into low-risk and high-risk groups, showing significantly poorer overall survival (OS) in the high-risk group. In conclusion, our risk assessment model exhibits strong potential for predicting poor survival and unfavorable clinicopathological features in CRC patients, offering valuable insights for personalised management strategies.

Introduction

Colorectal cancer (CRC) ranks as the third most commonly diagnosed malignancy and the second leading cause of cancer-related deaths globally, with over 1.9 million new cases and 935,000 deaths reported in 2020 (Sung et al., 2021). The majority of patients are diagnosed at an advanced stage, typically characterised by invasion, lymph node involvement, and distant metastasis, at which point treatment options remain limited (Wang et al., 2016; Woolf, 2000). Despite advances in surgical techniques, chemotherapeutic options, and the identification of several molecular biomarkers such as KRAS, BRAF mutations, and MSI status over the past decades, their predictive value remains limited and highlighting the need for novel prognostic markers (Koncina et al., 2020).

High-throughput global gene expression analysis technologies, such as microarray and RNA sequencing, offer valuable approaches for identifying potential prognostic markers related to survival and personalised CRC treatment (Chen et al., 2017; Kohne, 2014; Zuo, Dai & Ren, 2019). In this context, reported dysregulated gene expression-discovered biomarkers present significant potential for predicting clinical outcomes and classifying different subtypes (Li et al., 2017; Reitsam et al., 2024; Zhao et al., 2024). Additionally, there are various commercially available gene-expression-based multi-gene assays with prognostic value for CRC patients, such as Oncotype DX, ColoPrint, and ColDX (Fontana et al., 2019). The Oncotype DX® Colon Cancer Assay is one of the most popular tests for measuring the expression signatures of 12 genes in stage II colon cancer patients to predict the risk of recurrence (Fontana et al., 2019; Qian et al., 2021; Sun et al., 2018). However, these predictive factors remain inadequate as they provide only a limited reflection of the overall status of cancer patients. Given CRC’s complex pathogenesis and high heterogeneity, accurate patient stratification and the development of effective targeted therapies continue to pose challenges (Grothey, Fakih & Tabernero, 2021; Hull et al., 2020). An integrated strategy that identifies key biological pathways, control elements, and interactions within the multi-modular biology of cancer may lead to more robust, reproducible, and accurate predictions (Song et al., 2014). Therefore, further risk stratification is crucial to identify patients at high risk for advanced disease and to guide prognosis, management, and enable personalised therapy. In this context, the tumour microenvironment (TME), as a central component in the dynamics of cancer development and treatment, has attracted increasing attention. A key component within this microenvironment, the extracellular matrix (ECM), acts as a highly active and dynamic entity, significantly impacting tumour cell behaviour and metastatic capabilities (Karlsson & Nystrom, 2022). Throughout tumourigenesis, the ECM undergoes continuous remodelling during tumour formation, establishing a microenvironment that promotes tumour progression, invasion, angiogenesis, immune evasion, and drug resistance (Prakash & Shaked, 2024). In the context of colorectal cancer, the composition and structural arrangement of the ECM have been linked to tumour progression, patient outcomes, and responses to cancer treatments. Consequently, ECM components present a promising avenue for discovering novel prognostic markers and developing therapeutic strategies, as well as for improving risk stratification in patients (Karlsson & Nystrom, 2022; Yang et al., 2024).

In this study, we performed transcriptome profiling on forty-nine formalin-fixed paraffin-embedded (FFPE) matched tumour and normal tissue samples from sporadic colorectal cancer cases. Using an integrated approach, we combined our data with publicly available mRNA and miRNA datasets to identify candidate prognostic biomarkers. This analysis revealed four overexpressed genes (THBS2, FN1, COL1A1, and COL5A1), which were subsequently validated in an independently collected CRC cohort. We developed a four-gene prognostic signature and validated this model in two independent microarray datasets to predict overall survival (OS) in CRC. Our findings indicate that these genes are linked to CRC pathogenesis and that our risk score offers a novel prognostic biomarker.

Materials and Methods

Patient cohorts and sample preparations

A total of 49 matched tumour and normal FFPE sporadic colorectal cancer tissue samples (discovery dataset, Fig. 1: IIA1) obtained from the Ankara University, Faculty of Medicine, were used for transcriptome profiling. A second cohort of 64 tumours and matched normal CRC samples (validation dataset, Fig. 1: IIA2) was subsequently used to validate and test the reproducibility of the expression results. Table 1 presents the demographic and clinical features of participants in both the discovery and validation cohorts. Ethical approval for both the discovery (Ref: 153-4854) and validation (Ref: 15-200-19) cohorts was obtained from the Ankara University School of Medicine Clinical Research Ethics Committee. All participants provided informed consent, and the study was conducted in accordance with the Declaration of Helsinki.

Figure 1 Workflow of the current study.

We collected 49 paired tumour-control FFPE samples (98 samples in total) and conducted a transcriptome profiling analysis. Significantly differentially expressed genes were identified, and functional pathway enrichment analysis was performed. Then, an integrated mRNA microarray analysis was conducted using GEO CRC datasets. In both analyses, differentially expressed genes were primarily implicated in the ECM-receptor interaction pathway. To restrict the candidate gene list, we conducted an integrative meta-analysis for selected publicly available microRNA microarray datasets to identify miRNA-regulated genes in the ECM-receptor interaction pathway. Combining our data with the external mRNA/miRNA microarray analysis results, we identified four overexpressed genes in the ECM-receptor interaction pathway—a key regulator in human CRC development and progression. QRT-PCR validated the expression changes of the critical genes in the discovery microarray sample set and an independent cohort (64 tumours and 64 matched controls). The predictive value of our candidate genes was verified using the TCGA and independent GEO datasets and the high expression of all four genes correlated with poor CRC prognosis. A prognostic risk score was established for each patient by combining the expression values of the critical genes and disease stage weighted by regression coefficients in the multivariate Cox regression analysis. The reliability of the four-gene signature was validated using two independent GEO datasets (GSE17536 and GSE40967).

Table 1 Clinicopathological characteristics of patients included in the discovery and validation datasets.

	Discovery cohort	Validation cohort	
Characteristics	n (49)	%	n (64)	%	
Mean age (years)	45		67		
Sex					
Female	22	44.9	34	42.5	
Male	27	55.1	46	57.5	
Stage					
I	4	8.16	–	–	
II	21	42.86	36	45	
III	23	46.94	38	47.5	
IV	1	2.04	6	7.5	
Grade					
I	3	6.12	1	1.25	
II	26	53.06	66	82.5	
III	20	40.82	10	12.5	
Unknown	–	–	3	3.75	
Tumor location				
Left-sided	38	77.55	55	68.75	
Right-sided	11	22.45	25	31.25	

The FFPE tissue samples were fixed in 10% neutral-buffered formalin immediately after collection, embedded in paraffin blocks. The blocks used in discovery and validation datasets were 2–8 and 4–10 years old, respectively. A pathologist reviewed H&E slides to select sections containing ~90% tumour cells. Four 8-μm sections per block were cut, placed on slides (2 per slide), one stained with H&E to mark tumour areas and stored at −20 °C until used for transcriptome profiling analysis.

FFPE RNA extraction and microarray transcriptome profiling

The marked tumour regions were macrodissected from serial sections and used for RNA extraction with the RNeasy FFPE kit (Qiagen, Hilden, Germany), following a modified deparaffinization step as previously described (Belder et al., 2016). Genomic DNA was removed using the gDNA Eliminator spin column included in the kit. RNA concentration and purity were assessed with a NanoDrop ND-1000 Spectrophotometer (NanoDrop Technologies, Wilmington, DE, USA), and integrity was evaluated with an Agilent Bioanalyzer 2100 using the RNA 6000 Nano Assay (Agilent Technologies, Santa Clara, CA, USA).

According to the manufacturer’s instructions, 100 ng of input RNA was used to generate the amplified RiboSPIA product using the Ovation® FFPE WTA System (NuGEN, San Carlos, CA, USA). The fragmented and labelled samples (3.5 μg) were then hybridised to Affymetrix GeneChip® Human X3P Arrays (Affymetrix, Santa Clara, CA, USA) with rotation for 19 h at 45 °C. Subsequently, the arrays were washed and stained using user-prepared wash and staining reagents in a Fluidics Station 450 (Affymetrix) and scanned using a GeneChip Scanner 3000 (Affymetrix).

Microarray data pre-processing and analysis

The Affymetrix CEL files were pre-processed with Genomics Suite Version 6.6 (Partek Inc., St. Louis, MO, USA) using the robust multi-array analysis (RMA) algorithm, including background correction, quantile normalisation, logging of probes using base 2, and probe summarisation (Irizarry et al., 2003). A paired t-test was performed to determine which genes were differentially expressed between the tumour and matched normal samples, followed by Benjamini–Hochberg correction. The threshold for identifying differentially expressed genes (DEGs) was set as a false discovery rate (FDR) of <0.001 and |log2FC| ≥1. Unsupervised hierarchical cluster analysis was performed in the Partek Genomic Suite using the Euclidean distance and average linkage algorithm. The gene expression data were deposited in the NCBI’s Gene Expression Omnibus Database (GEO: GSE271719).

Pathway and gene set enrichment analysis

The Database for Annotation, Visualization and Integrated Discovery (DAVID, https://davidbioinformatics.nih.gov/) (Huang da, Sherman & Lempicki, 2009) was used to perform KEGG pathway enrichment analysis of the DEGs. A threshold-adjusted p-value of ≤ 0.05 was considered statistically significant. Gene Set Enrichment Analysis (GSEA) software (version 4.1.0) (http://software.broadinstitute.org/gsea/index.jsp) was employed to validate the ECM-receptor interaction pathway enrichment in CRC using our discovery dataset. The number of gene set permutations was set at 1,000. NES (normalized enrichment score) >1 and FDR (false discovery rate) q-value ≤ 0.05 were defined as the significant cut-off criterias of enrichment.

Integrated mRNA microarray dataset analysis

Differential gene expression and subsequent pathway enrichment analysis were conducted using independent colorectal cancer gene expression datasets to support our microarray data and pathway analysis. Three independent datasets (GSE9348, GSE24550, and GSE21510) were downloaded from the NCBI GEO repository (https://www.ncbi.nlm.nih.gov/geo/) to identify DEGs between tumour and normal samples for the integrated microarray dataset analysis (Fig. 1: I). Detailed information about the datasets, including the platform and the number of samples, is provided in Table S1. A threshold of |log2FC| ≥ 1 and an adjusted p-value of <0.05 were set to screen the DEGs. To analyse overlapping DEGs, we used an online Venn diagram tool (https://bioinfogp.cnb.csic.es/tools/venny/). KEGG pathway enrichment analysis was performed on the common DEGs using the DAVID bioinformatics resources.

miRNA microarray data set analysis

An integrative meta-analysis of publicly available miRNA microarray datasets was conducted to identify the miRNA-regulated genes involved in the ECM-receptor interaction pathway. Our goal was to narrow down the candidate gene list from our transcriptome analysis and identify key molecules relevant to CRC pathogenesis for further investigation.

We analysed a total of 161 samples (76 controls and 85 tumours) from the following datasets: NCBI GEO GSE68377 and GSE35982; Array Express E-MTAB-752, E-GEOD-35834, and E-MTAB-813. Detailed information on these datasets is provided in Table S2 (Fig. 1: III). Differential expression of miRNAs (DEMs) was assessed for each platform using cut-off values of adjusted p ≤ 0.05 and |FC| ≥1.5. DEMs were categorized as up-regulated or down-regulated for further analysis.

miRNA-mRNA target interactions

Potential target genes of DEMs were predicted using the miRTarBase (http://mirtarbase.mbc.nctu.edu.tw/php/index.php) and miRWalk2.0 (http://zmf.umm.uni-heidelberg.de/apps/zmf/mirwalk2/) databases. Overlapping analysis was performed using a Venn diagram to obtain a comprehensive list of miRNA target genes. Common genes were then identified by intersecting the miRNA target genes with the DEGs from our discovery dataset for further analysis. miRNA-mRNA pairs showing inverse correlation expression trends were filtered out, and genes involved in the ECM-receptor interaction pathway were selected as key genes from miRNA-mRNA target data analysis.

Determination of candidate target genes

Functional pathway enrichment analysis of our CRC gene expression microarray data and integrated microarray analysis of independent GEO CRC gene expression datasets revealed that the DEGs were primarily implicated in the ECM-receptor interaction pathway. We, therefore, focused on this pathway using an integrated approach to identify candidate genes more accurately by combining genes in the ECM pathway from our microarray study, the GEO mRNA datasets, and critical genes mentioned in the miRNA meta-analysis. The Venn diagram tool (http://bioinfogp.cnb.csic.es/tools/venny) was used to analyse overlapping genes. Five genes (THBS2, FN1, COL1A1, COL5A1, CD44) intersecting in the ECM-receptor interaction pathway were identified as candidate genes for further analysis. Univariate Cox regression analysis was applied to examine their association and patient overall survival (OS) using the GSE17536 and TCGA datasets (Fig. 1: II-2). Four of these genes (THBS2, FN1, COL1A1, and COL5A1), which showed a statistically significant association (p < 0.05) with patient OS, were identified as the critical target genes for subsequent analysis.

Quantitative real-time PCR (qRT-PCR)

To validate expression changes in the four target genes, qRT-PCR was conducted using our discovery dataset of RNA samples (n = 49). Retrospective validation was also performed using the validation cohort, which consisted of independently collected FFPE colorectal cancer samples and matched controls (n = 64). cDNA was synthesized from 0.5 μg total RNA using the Transcriptor First Strand cDNA Synthesis Kit (Roche, Basel, Switzerland) with random primers in a total volume of 20 μl. The reverse transcription reaction was carried out at 55 °C for 30 min, followed by 85 °C for 5 min. qRT-PCR was performed in triplicate using the SYBR Green I Master Kit (Roche, Basel, Switzerland) on a LightCycler480 instrument (Roche, Basel, Switzerland), following the manufacturer’s instructions. Amplification was performed at 95 °C/180 s followed by 40 cycles at 95 °C/30 s and 60 °C/30 s, 72 °C/30 s. Gene expressions were normalized to the geometric mean of HPRT and B2M mRNA levels. Ct values were acquired using the LightCycler® 480 software, and data were analyzed using the ΔΔCT method (Livak & Schmittgen, 2001). Primer sequences are listed in Table S3.

Verification of expression patterns and prognostic value of four candidate key genes in publicly available CRC datasets

External validation cohorts from the GEO and TCGA datasets were used to confirm the reproducibility of the overexpression of the four target genes in CRC (GSE44076, GSE24551, GSE39582, GSE32323, TCGA COAD-READ). Associations between elevated gene expression and advanced tumour progression (examined in GSE14333, GSE37892, GSE40967, GSE17538), as well as recurrence (examined in GSE40967, GSE17538), were also assessed, as shown in Fig. 1 (IIC1a). Statistical significance was set at p < 0.05. Additionally, the prognostic value of these genes in CRC (TCGA, GSE17538) was evaluated using Kaplan-Meier analysis, with p-values calculated via the log-rank test (GraphPad Prism 6.6). A log-rank p-value < 0.05 was considered statistically significant. Differences in gene expression for THBS2, FN1, COL1A1, and COL5A1 across different cancers were assessed using Oncomine (https://www.oncomine.org), with threshold parameters of p < 0.05, fold-change >2, and gene rank within the top 10%. To further investigate the potential biological roles of candidate prognostic genes in CRC, gene set enrichment analysis was performed using independent datasets (GSE17536 and GSE39582), as previously outlined. For each gene, patients were dichotomized into high- and low-expression groups based on the median expression value, and GSEA was carried out separately using KEGG pathway gene sets (c2.cp.kegg.v7.4.symbols.gmt).

Correlation analysis

The discovery dataset was subjected to Spearman correlation analysis in GraphPad Prism 6.6 to evaluate the interactions among the key genes at the transcription level. Correlation coefficients were calculated to assess the strength and direction of the correlations. Correlations among the genes were validated using external datasets, including GSE17536, GSE40967, and TCGA datasets.

Protein-protein interaction (PPI) network analysis

The online resource Search Tool for the Retrieval of Interacting Genes (STRING) database (http://stringdb.org/) was utilised to explore relationships for our crucial target proteins. After importing the candidate genes (THBS2, FN1, COL1A1 and COL5A1) into STRING, a confidence score of >0.4 was applied as the threshold for the PPI network information.

Determining the prognostic risk score for patient risk stratification

A multivariate Cox regression analysis was carried out on the discovery dataset with the log2 normalised expression values of the four candidate genes (THBS2, FN1, COL1A1, and COL5A1) and disease stage as the variables for each patient (p = 0.025). A combined gene score (∑(βi × Expi)) was calculated, where Expi represented the expression level of prognostic gene i, and βi was the regression coefficient for gene i obtained from the multivariate Cox analysis (Fig. 1: IIC2). The cohort was proportionally split into Low- and High- score values across the mean. The method was similarly applied to the GSE datasets and overall survival of the patients with high- and how- risk scores was compared by Kaplan-Meier survival and log-rank (Mantel-Cox) test to assess significance (Fig. 1: IIC2a).

Statistical analyses

All statistical analyses were conducted using GraphPad Prism 6.6 (GraphPad Software, Inc., San Diego, CA, United States), IBM SPSS® Statistics (version 20.0; IBM Company, New York, NY, USA) software package, and R v. 3.4.3. Univariate and multivariate Cox regression analyses were conducted to evaluate the predictive value of the target genes, risk scores, and clinicopathological features in CRC. All tests were two-tailed, and p-values of <0.05 were considered statistically significant.

Results

A combined prognostic model was established using an integrative approach incorporating the expression values of four genes (THBS2, FN1, COL1A1, and COL5A1) and each patient’s CRC disease stage. The risk model demonstrated independent prognostic value in CRC. The overall design of the present study is shown in Fig. 1.

Gene expression profile associated with sporadic colorectal cancer

Affymetrix GeneChip Human Genome U133 X3P expression arrays were used to identify differences in gene expression between the 49 cases of sporadic CRC FFPE tissue and their paired normal tissue samples. Applying the criteria of an adjusted p-value of <0.001 and a fold change (|log2FC|) of ≥1, a total of 845 DEGs were identified, comprising 320 up-regulated genes and 525 down-regulated genes (Table 2, Table S4). The distribution of the DEGs was visualised using the volcano plot in Fig. 2A, in which genes meeting the criteria of a p-value < 0.001 and |log2FC| ≥ 1 are represented by green and red dots, respectively. The PCA plot (Fig. 2B) shows a distinct separation between the tumour and control samples. Additionally, hierarchical clustering of the 845 DEGs across the 49 paired samples (Fig. 2C) demonstrated that DEG expression patterns effectively distinguished tumour tissues from normal counterparts.

Table 2 Number of differentially expressed genes.

Genes	Tumor vs. Control	
Up-regulated	320	
Down-regulated	525	
Total	845	

Figure 2 Heatmap, PCA, and volcano plot of significantly differentially expressed genes.

(A) Volcano plot of differentially expressed genes. Red points represent high expression genes; green points represent low expression genes; black points represent genes with no significant difference (FDR ≤ 0.001, |logFC| ≥ 1). (B) PCA scatter plot of CRC gene expression data. Each point represents a sample coloured by group status. Red dots represent tumour samples; blue dots represent normal samples. (C) Heatmap of significantly differentially expressed genes. Samples are indicated along the horizontal axis and grouped by the colour bar between the dendrogram and the heat map. Red indicates relatively high gene expression; green indicates relatively low gene expression; black indicates no significant changes in gene expression (FDR ≤ 0.001, |logFC| ≥ 1).

Table 3 lists the top 10 upregulated and downregulated DEGs. Among these, THBS2 (Thrombospondin-2), a member of the thrombospondin family, exhibited the highest upregulation (fold change = 20.20) in CRC compared to matched controls. The most downregulated gene (fold change = −39.40) was SLC26A3 (solute carrier family 26, member 3), which functions as an anion exchanger.

Table 3 Top ten up-regulated and down-regulated genes.

Gene symbol	Gene title	FDR	FC	
Up-regulated	
THBS2	Thrombospondin 2	1.40E−13	20.2049	
COL10A1	Collagen, type X, alpha 1	1.00E−11	18.735	
FAP	Fibroblast activation protein, alpha	8.44E−15	17.5485	
INHBA	Inhibin, beta A	8.08E−15	16.7212	
GRIN2D	Glutamate receptor, ionotropic, N-methyl D-aspartate 2D	1.70E−13	15.0628	
MMP11	Matrix metallopeptidase 11 (stromelysin 3)	5.84E−11	12.4528	
SULF1	sulfatase 1	1.13E−13	10.8833	
BGN	Biglycan	1.58E−14	10.7824	
SPP1	Secreted phosphoprotein 1	3.49E−09	9.76149	
H19	H19, imprinted maternally expressed transcript	2.00E−11	9.62265	
Down-regulated	
SLC26A3	Solute carrier family 26 member 3	8.28E−15	−39.4028	
CLCA1	Chloride channel accessory 1	1.16E−13	−36.9304	
CA2	Carbonic anhydrase II	4.17E−16	−34.482	
SLC4A4	Solute carrier family 4, member 4	3.76E−16	−34.0366	
CLCA4	Chloride channel accessory 4	2.57E−13	−33.4777	
MS4A12	Membrane-spanning 4-domains, subfamily A, member 12	2.81E-15	−32.2239	
CEACAM7	Carcinoembryonic antigen-related cell adhesion molecule 7	7.87E−14	−21.9921	
CHGA	Chromogranin A (parathyroid secretory protein 1)	1.05E−15	−20.2786	
CA1	Carbonic anhydrase I	2.60E−15	−19.9087	
SCNN1B	Sodium channel, non-voltage-gated 1, beta subunit	4.46E−15	−17.3185	

DEGs are primarily involved in the ECM-receptor interaction pathway

To explore the functional relevance of the 845 DEGs, KEGG pathway enrichment analysis was performed using the DAVID (v6.8) online tool and identified 13 enriched pathways (FDR < 0.05). The ECM-receptor interaction pathway was the most dysregulated in CRC, followed by pathways in cancer, focal adhesion, and PI3K-Akt signalling (Fig. 3A). Most upregulated genes were linked to focal adhesion and ECM-receptor interaction, while downregulated genes were mainly associated with metabolic processes (Fig. 3B). GSEA further confirmed significant enrichment of ECM-receptor signalling in tumour samples (Fig. 3C).

Figure 3 ECM-receptor interaction is a crucial pathway involved in colorectal cancer.

(A) Kyoto Encyclopaedia of Genes and Genomes Pathway (KEGG) enrichment analyses of the 845 DEGs from the discovery dataset. The bar graph presents the top deregulated pathways in CRC. The horizontal bars denote the different pathways based on the p values. Only results with p < 0.05 were considered significant. (B) Pathways of DEG in the discovery set based on gene count. The horizontal bars denote the different pathways based on the gene count in the pathways. (C) GSEA enrichment plot of ECM-receptor interaction pathway between tumour and paired normal samples from the discovery dataset. The red-to-blue colour bar shows the ranking of the genes of our microarray analysis for 49 paired tumour-normal samples from upregulated to downregulated in CRC. The vertical black bars present the position of the genes across the ranked gene list. The green line shows the running enrichment score (ES) across the ranked gene list. NES is the normalised enrichment score. Genes associated with ECM-receptor interaction are enriched in tumours from the discovery dataset (D) KEGG pathway enrichment analyses of the common genes in the integrated microarray analysis performed on three independent CRC GEO datasets. The bar graph presents the top deregulated pathways in CRC. The horizontal bars denote the different pathways based on the p-value. Only results with p < 0.05 were considered significant.

To support our findings, three independent CRC microarray datasets (GSE9348, GSE24550, GSE21510) from the NCBI GEO repository were analysed (Fig. 1: I)., resulting in 4,874, 5,634, and 7,853 DEGs (adjusted p < 0.05 and |log2FC| > 1), respectively (Table S5). The volcano plots for the DEGs in all three datasets are displayed in Figs. S1A–S1C. A Venn diagram revealed 971 common DEGs across these datasets (Fig. S2). Pathway analysis of these common DEGs highlighted four significant pathways in CRC: cell cycle, DNA replication, ECM-receptor interaction, and small-cell lung cancer (Fig. 3D). Considering both the discovery dataset and the external GEO CRC datasets for the pathway enrichment analysis, only the ECM-receptor interaction pathway emerged as a common and significantly enriched pathway. Given its crucial role in CRC, this pathway was selected as the starting point for identifying target genes and conducting further analyses.

THBS2, FN1, COL1A1, and COL5A1 were identified as candidate target genes

We conducted an integrative meta-analysis of selected publicly available miRNA microarray datasets to narrow down the enriched ECM-receptor interaction pathway gene list derived from the discovery dataset and integrated microarray analysis (Fig. 1: III). GSE68377 and GSE35982 (GEO datasets), E-MTAB-752, E-GEOD-35834, and E-MTAB-813 (Array Express datasets) (Table S2), were analysed separately. Differentially expressed miRNAs (DEMs) were identified for each platform (adj. p ≤ 0.05, |FC| ≥ 1.5) and classified as upregulated or downregulated miRNAs for further analysis. Overall, 110 miRNAs from the analysis of the Affymetrix E-MTAB-752, E-GEOD-35834, and GSE68377 datasets showed significant expression differences, with 63 upregulated and 47 downregulated. Additionally, we obtained 90 DEMs, including 75 upregulated and 15 downregulated miRNAs, from the analysis of the Agilent GSE35982 and E-MTAB-813 datasets (Table S6). Forty-two miRNAs were common across both platforms (Fig. S3).

Target genes for these forty-two miRNAs were predicted using miRTarBase and miRWalk2.0, and a Venn diagram identified overlapping genes with the DEGs from our discovery dataset (Table S7). miRNA-mRNA pairs showing an inverse correlation in expression trends were filtered out. Genes involved in the ECM-receptor interaction pathway and supported by miRNA analysis were prioritised (Fig. 4A). By intersecting ECM pathway genes from our microarray study, integrated analysis of GEO mRNA datasets, and miRNA analysis, we identified five candidate genes (THBS2, FN1, COL1A1, COL5A1, CD44) for further investigation (Fig. 1: II-1; Fig. 4B).

Figure 4 THBS2, FN1, COL1A1, and COL5A1 were identified as candidate target genes.

(A) The intersection of genes involved in the ECM-receptor interaction pathway in our discovery dataset and miRNAs in the integrated miRNA microarray analysis targeting these genes with inversely correlated expression trends. The red-to-blue colours show the ranking of the genes from upregulated to downregulated in CRC. Colour saturation correlates with gene level. (B) The Venn diagram depicts overlapping genes involved in the ECM-receptor interaction pathway between our discovery microarray study and integrated analysis of GEO mRNA datasets and miRNA meta-analysis. Colour images are available online. The five genes (THBS2, FN1, COL1A1, COL5A1, CD44) intersecting in the ECM-receptor interaction pathway were determined as key candidate genes for further analysis. (C, D) Forest plot of the five key genes based on a univariate Cox regression analysis in GSE17536 (C) and TCGA cohort (D). HR = Hazard ratio, 95%CI = 95% confidence interval. In the forest plot, statistically significant risk factors are depicted in red. (E) Heatmap of the key candidate genes. The expression of each gene in our discovery microarray dataset and three independent GEO CRC datasets used in integrated microarray analysis is shown in a coloured box with the log2 FC inset. Red and blue boxes indicate expression level, while colour saturation correlates with gene expression level. Thrombospondin 2 (THBS2), Fibronectin 1 (FN1), Collagen type I α1 chain (COL1A1), Collagen type I α1 chain (COL5A1), CD44 Molecule (Indian Blood Group) (CD44), The Cancer Genome Atlas (TCGA).

To assess their prognostic significance, a univariate Cox regression analysis was performed to examine the association between these five genes and patient OS using the GSE17536 dataset (Fig. 4C) and TCGA dataset (Fig. 4D). Among the five genes, four (THBS2, FN1, COL1A1, and COL5A1) showed a statistically significant association (p < 0.05) with patient OS and were therefore selected as essential target genes for subsequent analysis (Fig. 1: II-2). All four candidate genes were significantly upregulated (p < 0.05) in tumour samples compared to normal samples in both the discovery microarray dataset and the three independent GEO CRC datasets used for the integrated microarray analysis (Fig. 4E). The dysregulation and subsequent upregulation of these genes likely contribute to CRC tumorigenesis and prognosis.

THBS2, FN1, COL1A1, and COL5A1 gene expression changes in CRC were technically and externally verified

All four candidate genes (THBS2, FN1, COL1A1, COL5A1) were found to be upregulated in tumour samples compared to matched normal samples in our microarray analysis (Fig. 5A). To ensure the reliability of the differential expression of these four genes identified by our microarray profiling, we used qRT-PCR, the gold standard for confirming gene expression. The qRT-PCR results confirmed the upregulation of these genes in the same set of samples (p < 0.05) (Fig. 1: IIA1; Fig. 5B).

Figure 5 Validation of candidate target gene expression changes.

(A) Expression of THBS2, FN1, COL1A1, and COL5A1 in our discovery microarray dataset. (B) Up-regulation of the four genes was confirmed with qRT-PCR in the same set of our discovery samples. (C) Expression changes of our four key genes were externally validated in four independent GEO datasets (GSE44076, GSE24551, GSE39582, and GSE32323) and The Cancer Genome Atlas (TCGA) COAD-READ dataset—red, tumour, blue, normal. Data are shown as means ± SE. Differences between groups were analysed using Student’s t-test. *p < 0.05, **p < 0.01, and ***p < 0.001 indicate level of statistical difference from the control.

To assess the biomarker potential of these target genes, it is essential to validate their expression changes or upregulation in different and independent tumour samples. Therefore, we evaluated the robustness of the gene panel in four independent GEO colorectal cancer datasets (GSE44076, GSE24551, GSE39582, and GSE32323) and the TCGA COAD-READ dataset (Fig 1: IIA3). All four candidate genes were significantly upregulated in CRC patients compared to healthy controls, except for FN1 in the TCGA cohort (p < 0.05) (Fig. 5C). This validation across multiple and different cohorts highlights their potential importance in colorectal carcinogenesis.

Overexpression of THBS2, FN1, COL1A1, and COL5A1 in CRC tissues was further confirmed biologically

To validate the expression profile of the four candidate prognostic genes, a new set of independent cohort of 64 CRC tumours and matched normal samples was collected (Fig. 1 IIA2). The qRT-PCR results demonstrated that THBS2, FN1, COL1A1, and COL5A1 were significantly overexpressed in CRC tissues compared to their matched controls (p < 0.05) (Fig. 6A). Confirming the previously identified up-regulation in independent clinical samples highlights the validity and reliability of the results obtained from the microarray analysis. Furthermore, utilising the Oncomine database, we conducted a pan-cancer differential gene expression analysis for THBS2, FN1, COL1A1, and COL5A1 (Fig. 1 IIA3). As shown in Fig. 6B, our candidate genes were significantly upregulated in many cancer types compared to normal tissues. In particular, THBS2, COL1A1, and COL5A1 genes are upregulated in most cancer types, except for bladder, kidney, melanoma, and ovarian cancer. These results suggest that our essential genes may predominantly exhibit oncogenic characteristics in multiple types of cancer, particularly in CRC. To further investigate the functional roles of these genes in CRC progression, GSEA was performed using the GSE17536 and GSE39582 datasets. Patients were stratified into high- and low-expression groups based on the median expression values. Notably, the high-expression groups exhibited significant enrichment in several pathways, including focal adhesion, actin cytoskeleton regulation, and cell adhesion molecules (CAMs), in addition to the ECM-receptor interaction pathway. Furthermore, the KEGG mTOR signalling pathway was also enriched, suggesting a potential link between these genes and mTOR signalling in CRC progression. These findings offer a potential mechanistic link between the candidate genes and CRC progression (Tables S8–S15).

Figure 6 Independent validation in CRC and expression analysis in different types of cancers (ONCOMINE).

(A) qRT-PCR analysis of THBS2, FN1, COL1A1, and COL5A1 in independently collected colorectal cancer tissue samples. All four genes were significantly up-regulated in tumour tissues (p < 0.05) compared to matched normal tissues. Data are shown as means ± SE. Differences between groups were analysed using Student’s t-test. **p < 0.01, and ***p < 0.001 indicate level of statistical difference from the control. (B) Expression of THBS2, FN1, COL1A1, and COL5A1 in various types of cancer (ONCOMINE). Up-regulation is highlighted in red, while down-regulation is highlighted in blue. The number in each cell displays the number of unique analyses that met the given cut-offs. Colour intensity indicates the best rank of the gene in the analyses. Cut-off p values and fold changes were as follows: p-value: 0.05, fold change: 2, gene rank: 10%. Cell colour gradients represent the gene rank.

High expression of THBS2, FN1, COL1A1 and COL5A1 is associated with advanced stage and worse prognosis in CRC patients

We analysed the mRNA expression of THBS2, FN1, COL1A1, and COL5A1 across CRC stages using four GEO datasets (GSE14333, GSE17538, GSE40967, GSE37892) (Fig. 1 IIC1a). A significant increase in expression was observed from early to late stages (p < 0.05), suggesting their role in CRC progression (Figs. 7A–7D). We then examined whether the overexpression of the four genes was associated with the prognosis of the CRC patients in the GSE17538 and TCGA CRC datasets (Fig. 1 IIC1b). The results indicated that high expression of THBS2 (p = 0.0016, p = 0.018), FN1 (p < 0.0001, p = 0.0532), COL1A1 (p = 0.018, p = 0.0239), and COL5A1 (p = 0.0001, p = 0.067) was significantly associated with poor OS in GSE17538 and TCGA (Figs. 7E, 7F). Additionally, these genes were significantly upregulated in the recurrent CRC group compared to the non-recurrent group, further highlighting their prognostic relevance. (p < 0.05) (Figs. 7G, 7H). Spearman’s correlation analysis results revealed clear positive correlations (r > 0.6, p < 0.001) between genes in colorectal cancer in the discovery dataset (Fig. 1: IIB; Fig. 8A), which were confirmed in the independent GEO and TCGA datasets (GSE17536: r > 0.6, p < 0.001; GSE40967: r > 0.6, p < 0.001; TCGA COAD-READ: r > 0.8, p < 0.001) (Figs. 8B–8D). A heatmap further visualized these co-expressions, highlighting their collaborative role in CRC (Figs. 8E–8H). Notably, the strongest positive correlation across all datasets was observed between COL1A1 and COL5A1 (discovery dataset: r = 0.86, p < 0.0001; GSE17536: r = 0.95, p < 0.0001; GSE40967: r = 0.96, p < 0.0001; TCGA COAD-READ: r = 0.97, p < 0.0001). Finally, a PPI network analysis in the STRING confirmed a highly significant interactions, highlighting the biological interconnectivity of these genes (PPI enrichment p = 1.15e−05) (Fig. 8I).

Figure 7 Prognostic significance of the identified candidate genes THBS2, FN1, COL1A1, and COL5A1 in colorectal cancer.

(A–D) mRNA expression levels of the four candidate genes across different cancer stages. Expression levels were correlated with pathological T stages or ductal stages. *p < 0.05, **p < 0.01, ***p < 0.001. (E–H) Prognostic significance of THBS2, FN1, COL1A1, and COL5A1 in colorectal cancer. Overall survival analysis was performed by stratifying patients into high- and low-expression groups based on the median log2 median-centred intensity values. The p-value was calculated from the log-rank test using GraphPad Prism 6.6 *p < 0.05, **p < 0.01, ***p < 0.001.

Figure 8 Co-expression and interaction analysis of THBS2, FN1, COL1A1, and COL5A1 in CRC.

(A–D) Spearman’s correlation of mRNA expression levels for THBS2, FN1, COL1A1, and COL5A1 genes in CRC across different datasets: (A) Discovery dataset; (B) GSE17536; (C) GSE40967; (D) TCGA. (E) Spearman’s correlation heatmaps of gene expression in the same datasets, with correlation coefficients visualized using a continuous color gradient. Red indicates a positive correlation, while blue indicates a negative correlation. (F) Protein–protein interaction network of the key target genes based on the STRING database.

A four-gene signature-based risk assessment model was established to predict the prognosis of colorectal cancer

Given the limitations of using a single gene to predict prognosis in a complex disease like cancer, we evaluated the combined prognostic value of the four genes. Since tumour stage is a crucial factor in patient survival (Brierley et al., 2019), we developed a prognostic risk score model that incorporates the expression levels and regression coefficients of these four genes alongside tumour stage to predict overall survival (OS) in our discovery dataset (Fig 1: IIC2). We then calculated the risk score for each patient and divided the CRC patients into high-risk and low-risk groups using the median cut-off of the risk score. Kaplan-Meier plots indicated significant differences in OS between the two groups. Patients in the high-risk group showed significantly poorer OS compared to those in the low-risk group (p = 0.0443; Fig. 9A).

Figure 9 Predictive values of the four-gene signature-based risk score model.

Kaplan–Meier curve of the risk score for OS in (A) Discovery dataset; (B) GSE17536; (C) GSE40967. The p values were calculated from the Log-rank test using GraphPad Prism 6.6. Violin plot visualisation of the expression levels of the four genes in the high- and low-risk groups in (D) Discovery dataset, (E) GSE17536, and (F) GSE490967. Data are shown as means ± SE. Differences between groups were analysed using Student’s t-test. ***p < 0.001. Heatmap plot of the four signature genes in each sample with different risk scores in (G) the Discovery cohort; (H) GSE17536; (I) GSE40967. Rows represent genes; columns represent patients. The colour spectrum from blue to red represents low to high gene expression.

To assess the robustness and effectiveness of our risk score model and verify its reproducibility, we used two independent GEO datasets (GSE17536, n = 145 and GSE40967, n = 560) with OS and gene expression information of the CRC patients (Fig 1. IIC2a). As shown by the Kaplan–Meier OS curves in Figs. 9B and 9C, patients in the high-risk group had shorter overall survival than those in the low-risk group (GSE17536: p < 0.0001; GSE40967: p = 0.0446) (Fig. 9B; Fig. 9C). Gene expression for all four prognostic genes was significantly higher in the high-risk group (p < 0.0001) (Figs. 9D–9I). Overall, the results confirmed that our risk score model, based on the four-gene scoring system and incorporating disease stage, could provide an accurate prognosis for CRC patients.

Discussion

This study elucidates the molecular basis of colorectal cancer (CRC) prognosis by identifying gene signatures linked to overall survival, thereby enabling improved patient stratification and therapeutic decision-making. Through transcriptomic analysis of 49 matched tumour and normal tissue samples from sporadic CRC cases (n = 98), we identified differential gene expression, predominantly enriched in the ECM–receptor interaction pathway, a finding corroborated by Gene Set Enrichment Analysis. To validate the robustness of our microarray data and pathway analysis, we conducted an integrated mRNA expression analysis across three independent CRC datasets, uncovering 971 overlapping DEGs. Subsequent pathway analysis demonstrated enrichment in cancer related pathways; notably, only the ECM–receptor interaction pathway was consistently enriched across both discovery and validation cohorts, underscoring its recurrent biological relevance in CRC.

ECM’s abundance, composition, and structural alterations, along with the resulting cell-ECM interactions, have crucial implications for cellular and tissue functions (Winkler et al., 2020). Alterations in ECM composition and structure profoundly affect tumour biology, enabling hallmark features such as uncontrolled proliferation, apoptosis resistance, angiogenesis, invasion, and immune evasion (Hanahan & Weinberg, 2011; Pickup, Mouw & Weaver, 2014; Winkler et al., 2020). Mounting evidence suggests that ECM remodelling not only underpins tumour progression but also fosters metastasis and therapeutic resistance across various cancers, including breast cancer (Bao et al., 2019), squamous cell carcinoma (Parker et al., 2022), pancreatic ductal adenocarcinoma (Laklai et al., 2016), and ovarian cancer (Januchowski et al., 2014). In addition to its direct effects on tumour cells, ECM remodelling critically influences the stromal compartment, shaping the tumour microenvironment (TME). In CRC, dynamic interactions between ECM components and stromal cells, particularly cancer-associated fibroblasts (CAFs), promote tumour invasion, metastasis, and therapy resistance (Henke, Nandigama & Ergun, 2019; Kim et al., 2021; Petersen et al., 2020). Recent studies have further emphasised the pivotal role of the stromal compartment in defining tumour phenotypes. Calon et al. (2015) demonstrated that poor-prognosis CRC subtypes are primarily defined by stromal gene expression, particularly TGF-β–induced programs in CAFs (Calon et al., 2015). Similarly, Isella et al. (2015) reported that a substantial proportion of the CRC transcriptome originates from stromal cells, correlating with therapeutic failure and adverse outcomes (Isella et al., 2015). Building on this, Lee et al. (2024) showed that dysadherin contributes to ECM remodelling through the MMP9 axis and CAF activation, promoting a more aggressive tumour microenvironment (Lee et al., 2024). Along with our findings, these observations highlight ECM remodeling and stromal dynamics as key contributors to an aggressive tumour microenvironment in CRC, with potential as both prognostic biomarkers and therapeutic targets.

Given the pivotal role of ECM–receptor interactions in colorectal cancer pathogenesis, we sought to refine our gene panel by integrating miRNA expression data to explore potential post-transcriptional regulatory mechanisms affecting ECM-related genes. miRNAs are well-known post-transcriptional regulators and are frequently dysregulated in cancer (Bracken, Scott & Goodall, 2016; Lin & Gregory, 2015). By analysing publicly available miRNA datasets, we identified differentially expressed miRNAs and predicted their target genes within the ECM–receptor interaction pathway. This integrative approach revealed five commonly overexpressed genes—THBS2, FN1, COL1A1, COL5A1, and CD44—involved in ECM–receptor interactions. We subsequently evaluated the the predictive value of these five genes for OS in the GSE17536 and TCGA CRC datasets. Four genes (THBS2, FN1, COL1A1, and COL5A1) demonstrated statistically significant associations with OS, indicating that patients with high expression of these genes have worse OS outcomes. Additionally, all four candidate genes showed significant upregulation in tumour samples compared to normal samples in our discovery microarray dataset, independent GEO CRC, and the TCGA COAD-READ dataset. Given the critical importance of independent validation for substantiating the reproducibility, generalizability, and biomarker potential of these findings (Ramspek et al., 2021; Shariat et al., 2010), we conducted a biologically independent validation using 64 archival FFPE samples comprising matched colorectal tumour and normal tissues. Quantitative RT-PCR analysis confirmed significant upregulation of all candidate genes in tumour samples. To further contextualize these results, we examined their expression profiles across multiple cancer types using Oncomine. Notably, THBS2, FN1, COL1A1, and COL5A1 were consistently overexpressed in colorectal and other malignancies compared to corresponding normal tissues, reinforcing their potential oncogenic roles in tumorigenesis. To gain additional mechanistic insight beyond expression patterns, we performed complementary in silico analyses using gene set enrichment analysis (GSEA) across two large, independent CRC cohorts (GSE17536 and GSE39582). These analyses revealed consistent enrichment of key oncogenic pathways, including ECM–receptor interaction, focal adhesion, actin cytoskeleton regulation, and notably, the mTOR signalling pathway, among high-expression groups of THBS2, FN1, COL1A1, and COL5A1. While the enrichment of ECM-related pathways is in line with the known roles of THBS2, FN1, COL1A1, and COL5A1 in extracellular matrix remodelling and tumour progression, the concurrent upregulation of mTOR signalling presents an intriguing finding. Recent evidence suggests a reciprocal relationship between ECM remodelling and mTOR pathway activation in cancer. Integrin-mediated signalling triggered by ECM components activates the PI3K/AKT/mTOR pathway and contributes to metastatic potential (Gargalionis et al., 2022; Matsuoka et al., 2012). In turn, mTOR activation has been shown to upregulate MMP-2 and MMP-9—downstream effectors that degrade the ECM and facilitate tumour cell invasion (Xu et al., 2023). This bidirectional interaction between ECM-integrin signalling and mTOR-MMP axis may underlie the coordinated activation of these pathways in CRC and highlights their mechanistic relevance in tumour progression. In colorectal cancer, ECM components such as collagen 1 have been shown to activate PI3K/AKT/mTOR signalling and promote stemness and metastasis (Wu et al., 2019). Furthermore, THBS2 appears to play a multifaceted role. In addition to promoting angiogenesis via PI3K/AKT activation (Huang et al., 2023), THBS2-expressing cancer-associated fibroblasts have been identified as drivers of oxaliplatin resistance through the secretion of COL8A1, which activates PI3K/AKT signalling and induces epithelial–mesenchymal transition (Zhou et al., 2024). Despite accumulating evidence of the interplay between ECM remodeling and mTOR signalling across various malignancies, studies specifically linking ECM-related genes to mTOR pathway activation in colorectal cancer remain limited. Our findings thus further substantiate the connection between ECM remodelling and colorectal cancer progression and provide a foundation for future hypothesis-driven studies.

We examined the interaction between candidate genes and pathological stages in CRC patients, and found that high THBS2, FN1, COL1A1, and COL5A1 expression levels were positively associated with disease progression. These genes were also significantly overexpressed in the recurrent group compared to the non-recurrent group, suggesting their role in CRC pathogenesis. Additionally, analysis in independent datasets (GSE17538 and TCGA COAD-READ) confirmed that elevated expression of these genes predicts poorer OS in CRC patients. In conclusion, our findings highlight the critical role played by our candidate genes in CRC and suggest that these genes individually may be therapeutic targets for CRC and valuable for prognostic and diagnostic purposes. These results are consistent with previous studies demonstrating the pathological significance of these genes in cancer progression. For instance, Deng, Liu & Wang (2021) reported that elevated THBS2 expression correlates with shorter overall and disease-free survival, alongside a positive association with immune infiltrates in CRC, supporting its role as both a prognostic and immunological biomarker. Similarly, Qu et al. (2022) reported a significant upregulation of THBS2 in CRC tissues compared to normal tissues. They found that THBS2 overexpression enhanced the migration and invasion of CRC cells and was associated with worse OS in CRC patients (Qu et al., 2022). A recent study demonstrated that THBS2 secreted by cancer-associated fibroblasts (CAFs) promotes CRC progression by binding to CD47 on tumour cells and activating the MAPK/ERK5 signalling cascade (Liu et al., 2025). In addition, Zhou et al. (2024) identified a distinct subset of THBS2-expressing CAFs as key drivers of oxaliplatin resistance, wherein COL8A1 secreted from these cells activates the PI3K-AKT signalling pathway, thereby inducing epithelial-mesenchymal transition and promoting chemoresistance in CRC. Collectively, these studies underscore the multifaceted role of THBS2 in CRC biology. Fibronectin 1 (FN1), a one of the major extracellular matrix glycoprotein, is known to mediate cell adhesion, migration, and invasion (Wang et al., 2022; Zhang et al., 2022). Numerous studies have shown that elevated FN1 expression is associated with the development and progression of various cancer types, including head and neck squamous cell carcinoma (Zhou et al., 2022), hepatocellular cancer (Zhang et al., 2022), thyroid cancer (Chen & Shen, 2023), gastric cancer (Li et al., 2022a), ovarian cancer (Lou et al., 2013) and renal cancer (Waalkes et al., 2010). Although the precise mechanisms underlying FN1-mediated tumorigenesis in CRC remain to be fully elucidated, its upregulation has been shown to promote tumour growth and confer drug resistance in CRC (Ye, Zhang & Feng, 2020). In line with this, FN1 has been shown to promote colorectal cancer progression by enhancing cell viability, invasion, and migration through interaction with ITGA5. Its elevated expression in CRC tissues correlates with adverse clinicopathological features and poorer survival outcomes, underscoring its potential as a prognostic biomarker and therapeutic target (Cai et al., 2018). The extracellular matrix is an essential component of the tumour microenvironment, with COL1A1 playing a crucial role in tumour development and metastasis as a significant structural ECM protein (Geng et al., 2021; Lu, Weaver & Werb, 2012). COL1A1 overexpression has been reported across several malignancies, including breast, gastric, and hepatocellular carcinoma (Geng et al., 2021; Li et al., 2022b), and is associated with poor prognosis, metastasis, and drug resistance (Li et al., 2022b). In CRC, Zhang et al. (2018a, 2018b) demonstrated that COL1A1 upregulation promotes tumour progression and metastasis, while its inhibition markedly suppresses cell proliferation, invasion, and migration. They also reported that elevated COL1A1 expression is associated with poorer outcomes and disease-free survival patients with CRC patients (Zhang et al., 2018a, 2018b). Collectively, the findings from these studies indicate that COL1A1 is a potential prognostic biomarker and a promising therapeutic target for CRC patients. COL5A1, the final gene in our prognostic panel, is upregulated across multiple malignancies, including glioma, clear cell renal cell carcinoma, and gastric cancer (Feng et al., 2019; Gu et al., 2021; Yang et al., 2022). Overexpression of COL5A1 is also associated with tumorigenicity and metastasis, supporting its potential as a prognostic biomarker and therapeutic target. Notably, Zhang et al. (2021) reported a correlation between COL5A1 overexpression and tumour-infiltrating immune cells (TIICs), proposing its utility in predicting ovarian cancer progression and paclitaxel (PTX) resistance (Zhang et al., 2021). However, to our knowledge, the clinical significance and role of COL5A1 in CRC are still unknown. Our study is the first to systematically evaluate the prognostic value of COL5A1 in CRC through integrated bioinformatics and biological analyses. Functionally related genes are frequently co-expressed (Eisen et al., 1998), and several studies have demonstrated that assessing gene co-expression can aid in identifying appropriate targets and uncovering regulators underlying diseases and other phenotypes (Shi, Derow & Zhang, 2010; Van Dam et al., 2018). Therefore, we conducted a Spearman’s correlation analysis better to understand the relationship among the four candidate genes. We found they were positively intercorrelated in CRC in the discovery, GSE17536, GSE40967, and TCGA COAD-READ datasets. We also examined the PPI network for our key target genes using the STRING dataset, revealing significant interactions between them, reinforcing their potential functional linkage in CRC pathophysiology.

Although conventional clinicopathologic features, such as TNM staging, currently aid in predicting CRC prognosis to some extent, these parameters often fail to biologically differentiate tumours, resulting in ineffective treatments (Eschrich et al., 2005; Feng et al., 2022; Sveen, Kopetz & Lothe, 2020). Given CRC’s heterogeneous and complex biology, there is a need to establish more accurate and valuable prognostic models. It is also important to note that combining prognostic models with conventional clinicopathological parameters offers better predictive potential than a single biomarker. Recently, gene expression-based risk assessment models have gained significant attention and shown enormous potential in predicting prognosis in various cancers, including CRC (Cheong et al., 2022; Liu et al., 2019; Zhou et al., 2020). In the current study, we evaluated the expression of the proposed key genes (THBS2, FN1, COL1A1, and COL5A1) in combination with the disease stage as a clinical parameter to develop a new prognostic prediction model. Survival analysis revealed that patients with high-risk scores had considerably shorter OS times than patients with low-risk scores. The predictive accuracy of our four-gene signature-based risk assessment model was further validated in two independent CRC gene expression datasets. While the model demonstrated modest but statistically significant performance in the discovery cohort (p = 0.0443), it was independently validated in two external datasets—GSE17536 (p < 0.0001) and GSE40967 (p = 0.0446). This dual validation across cohorts with distinct clinical backgrounds supports the robustness and potential generalisability of the ECM-associated risk score. We believe that our findings will provide a new and accurate classification and treatment strategy for CRC, which will improve patient outcomes. Nonetheless, despite consistent performance across retrospective cohorts, validation in large-scale, prospective clinical cohorts would be valuable for confirming the model’s translational relevance and facilitating its integration into routine clinical practice.

Despite the robust multi-tiered validation strategy employed in this study—including experimental, clinical and in silico analyses—we acknowledge that the relatively modest sample sizes in both the discovery and validation cohorts may limit the broader generalisability of our findings. To strengthen the translational potential of this gene set, future investigations involving larger, prospectively collected patient cohorts will be essential to confirm their prognostic value and clinical relevance. Moreover, comprehensive functional studies—encompassing protein-level validation and both in vitro and in vivo experiments—are required to elucidate the biological roles of these ECM-associated genes in colorectal cancer pathogenesis.

Conclusion

In conclusion, our findings underscore the critical role of our candidate genes in CRC, particularly in the context of the ECM-receptor interaction pathway, suggesting their potential as therapeutic targets and valuable biomarkers for prognosis and diagnosis. All these genes are overexpressed in a variety of tumours as well in CRC, and they may play crucial roles in tumour development and progression rather than being expressed merely as a result of the cancer cells phenotypic change. We also constructed and validated a new four-gene expression signature model in combination with cancer stage to divide CRC patients into poor- and good-prognosis groups that could contribute to overall survival prediction and management of individualized therapy. However, further research, including clinical validation of our findings will provide new insights into developing alternative therapeutic strategies and personalised therapies.

Supplemental Information

Supplemental Information 1 Supplemental tables.

Supplemental Information 2 Supplemental figures.

Supplemental Information 3 MIQE checklist.

We thank the Ozdag laboratory members for their invaluable discussion and advice.

Additional Information and Declarations

Competing Interests

Hilal Ozdag is an Academic Editor for PeerJ.

Author Contributions

Nevin Belder conceived and designed the experiments, performed the experiments, analyzed the data, prepared figures and/or tables, authored or reviewed drafts of the article, and approved the final draft.

Sulgun Charyyeva performed the experiments, analyzed the data, authored or reviewed drafts of the article, and approved the final draft.

Edibe Ece Abaci Oruc performed the experiments, analyzed the data, authored or reviewed drafts of the article, and approved the final draft.

Hakiimu Kawalya performed the experiments, analyzed the data, authored or reviewed drafts of the article, and approved the final draft.

Namood-e Sahar performed the experiments, analyzed the data, authored or reviewed drafts of the article, and approved the final draft.

Nader Omidvar performed the experiments, analyzed the data, authored or reviewed drafts of the article, and approved the final draft.

Berna Savas analyzed the data, authored or reviewed drafts of the article, contributed to the collection of FFPE tumour and normal samples, as well as the analysis and identification of tumour and normal regions, and approved the final draft.

Arzu Ensari analyzed the data, authored or reviewed drafts of the article, contributed to the collection of FFPE tumour and normal samples, as well as the analysis and identification of tumour and normal regions, and approved the final draft.

Hilal Ozdag conceived and designed the experiments, performed the experiments, analyzed the data, prepared figures and/or tables, authored or reviewed drafts of the article, and approved the final draft.

Human Ethics

The following information was supplied relating to ethical approvals (i.e., approving body and any reference numbers):

Ethical approval was obtained from the Ankara University School of Medicine Clinical Research Ethics Committee (Ref: 153-4854). Ethical approval for the second cohort was also obtained for the validation part of the study (Ref: 15-200-19).

DNA Deposition

The following information was supplied regarding the deposition of DNA sequences:

Our four candidate genes sequences are available at GenBank: THBS2: NM_003247.5, COL1A1: NM_000088.4, COL5A1: NM_000093.5, and FN1: NM_212482.4.

Data is available at: https://www.cbioportal.org/study/summary?id=coadread_tcga_pan_can_atlas_2018.

Microarray Data Deposition

The following information was supplied regarding the deposition of microarray data:

Data is available at the Gene Expression Omnibus: GSE271719.

Data Availability

The following information was supplied regarding data availability:

Gene expression data are available at the NCBI Gene Expression Omnibus database: GSE9348, GSE17536, GSE40967, GSE24550, GSE21510, GSE68377, GSE35982, GSE44076, GSE24551, GSE32323, GSE14333, GSE37892, and GSE17538. High-throughput functional genomic data is available from Array Express: E-MTAB-752, E-GEOD-35834, and E-MTAB-813.

The TCGA COAD-READ dataset used in this study is publicly available through The Cancer Genome Atlas (TCGA) portal

(https://portal.gdc.cancer.gov/projects/TCGA-COAD) and cBioPortal database (https://www.cbioportal.org/study/summary?id=coadread_tcga_pan_can_atlas_2018/).

The microarray data (discovery dataset) of colorectal cancer FFPE samples compared to paired normal samples are available at the Gene Expression Omnibus database: GSE271719.

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
