# Peer review of "Transformative insights from transcriptome analysis of colorectal cancer patient tissues: identification of four key prognostic genes"

_PeerJ, doi:10.7717/peerj.19852_

## Round 0.1 · original submission · Major Revisions

· Academic Editor

Major Revisions

The reviewers found your manuscript interesting, however they had a number of concerns that need to be addressed. One major concern is that the sample size for both the discovery and validation cohorts is small. The reviewers suggested that you address the limitations of having a small sample size in the manuscript itself and that you also use a larger cohort for additional validation, particularly since the performance of the risk score model in your discovery cohort is modest. Furthermore, functional validation is needed to establish disease causality with the genes identified in the prognostic signature, particularly for those that had not been previously implicated in other studies. Lastly, previous studies identifying ECM-related genes as prognostic need to be cited and included in the discussion.

Please, submit a detailed rebuttal which shows where and how you have taken all comments and suggestions into consideration. If you do not agree with some of the reviewers’ comments or suggestions, please explain why. Your rebuttal will be critical in making a final decision on your manuscript. Please, note also that your revised version may enter a new round of review by the same or by different reviewers. Therefore, I cannot guarantee that your manuscript will eventually be accepted.

·

Basic reporting

no comment

Experimental design

no comment

Validity of the findings

no comment

Additional comments

I have carefully read the manuscript entitled “Transformative insights from transcriptome analysis of colorectal cancer patient tissues: identification of four key prognostic genes” by Belder et al. This study identifies four key genes in the ECM-receptor interaction pathway that play critical roles in colorectal cancer development and progression. By developing a novel prognostic risk score model using multivariate Cox regression, the researchers demonstrate a promising approach to stratifying CRC patients into low-risk and high-risk groups, potentially offering personalized management strategies for improved clinical outcomes. However, I have some comments that need to be addressed before the manuscript can be accepted for publication.

Major comments:
1. The manuscript is generally well-written in professional English, though minor grammatical errors and occasional awkward phrasing are present. A thorough proofread by a native English speaker or professional editing service is recommended to enhance readability for an international audience.
2. The introduction effectively contextualizes the study, highlighting the clinical significance of CRC and the limitations of existing prognostic biomarkers. Literature citations are relevant and up-to-date. However, the rationale for focusing on the ECM-receptor interaction pathway could be strengthened by discussing prior evidence linking this pathway to CRC progression.
3. The pictures are generally blurry.
4. Metadata for supplementary files (e.g., clinical characteristics in Table 1) lack sufficient detail (TNM staging, pathology, differentiation, radiotherapy or not). Provide descriptive filenames and annotations for all supplementary data.
5. The discovery cohort (n=49) and validation cohort (n=64) are relatively small. While external validation using GEO/TCGA datasets mitigates this, acknowledge sample size limitations in the Discussion.
6. The risk score model’s performance in the discovery cohort (p=0.0443) is modest. The stronger validation in GSE17536 (p<0.0001) supports its utility, but further validation in prospective cohorts is needed.
7. The focus on ECM-receptor interaction aligns with CRC biology, and the association of THBS2, FN1, COL1A1, and COL5A1 with poor prognosis is compelling. However, external biological validation and functional experiments (e.g., knockdown/overexpression in vitro/in vivo) are lacking to establish causality, this is going to be the biggest hard-hitting part of the whole article.

The manuscript would benefit from additional quantitative data and a more detailed information. I hope that the authors will consider my comments and suggestions seriously and revise their manuscript accordingly. I look forward to seeing an improved version of their manuscript soon.

Reviewer 2 ·

Basic reporting

The four identified genes were independently validated for differential expression and prognostic association in a newly collected CRC cohort and publicly available datasets. However, further validation in larger, multi-cohort studies is needed.
1) How does the small sample size (49 matched samples) impact the statistical power and reproducibility of the identified prognostic biomarkers?
• Given the high-dimensional nature of gene expression data, small sample sizes can lead to overfitting and false-positive discoveries. How did the study mitigate these risks, and was an independent large-cohort validation performed to confirm the robustness of the findings?
2) Were the identified four genes functionally validated to confirm their causal role in CRC progression?
• While differential expression and statistical associations suggest a potential prognostic role, have functional studies (e.g., knockdown/overexpression experiments) been conducted to establish their direct involvement in CRC progression and therapeutic relevance?

Experimental design

see above

Validity of the findings

see above

Additional comments

see above

Reviewer 3 ·

Basic reporting

1. The font size in Figures 1 and 2 needs to be increased to ensure clarity and readability, which is essential for professional scientific communication.

2. The manuscript requires additional literature review and context, particularly:

- The discussion of ECM-related genes and their prognostic significance should be expanded by incorporating PMID 25706628 and 25706627 findings.

- More comprehensive discussion and citations are needed regarding previous THBS2, FN1, and COL1A1 studies.

Experimental design

3. The study demonstrates originality in identifying COL5A1 as a novel prognostic factor in colorectal cancer.

4. The research objective is clearly defined, focusing on identifying prognostic factors in colorectal cancer through the analysis of ECM regulatory genes.

5. Optional consideration: If the dataset represents a specific ethnic population, highlighting this aspect could enhance the study's context and relevance.

Validity of the findings

6. The overall analysis methodology appears robust and technically sound.

7. The data support the findings regarding the integrated analysis of THBS2, FN1, COL1A1, and COL5A1.

8. The conclusions appropriately connect to the original research question, particularly in demonstrating the importance of these genes in patient prognosis prediction.

---

## Round 0.2 · accepted · Accept

· Academic Editor

Accept

Thank you for thoroughly addressing the reviewers' comments and thus greatly improving your manuscript.

Reviewer 3 ·

Basic reporting

The authors have adequately addressed all the concerns raised in my previous review. The figures have been improved with better resolution and readability. The literature review has been expanded appropriately, incorporating the suggested references (PMID 25706628 and 25706627) and providing more comprehensive discussion of THBS2, FN1, and COL1A1. The manuscript now meets the standards for basic reporting.

Experimental design

no comment

Validity of the findings

The authors have provided satisfactory responses to the concerns about sample size limitations and functional validation. While in vitro/in vivo experiments would strengthen the findings, the comprehensive validation strategy using multiple independent datasets and the addition of GSEA analysis provide reasonable support for their conclusions. The findings are valid within the scope of the current study.

Additional comments

no comment